# Identifying critical features of iron phosphate particle for lithium preference

Gangbin Yan [1], Jialiang Wei[2], Emory Apodaca[1], Suin Choi[1], Peter J. Eng[3,4], Joanne E. Stubbs [3], Yu Han [1], Siqi Zou[1], Mrinal K. Bera[5], Ronghui Wu [1], Evguenia Karapetrova[6], Hua Zhou [6], Wei Chen [2,7] & Chong Liu [1] ✉

One-dimensional (1D) olivine iron phosphate ($FePO_4$) is widely proposed for electrochemical lithium (Li) extraction from dilute water sources, however, significant variations in Li selectivity were observed for particles with different physical attributes. Understanding how particle features influence Li and sodium (Na) co-intercalation is crucial for system design and enhancing Li selectivity. Here, we investigate a series of $FePO_4$ particles with various features and revealed the importance of harnessing kinetic and chemomechanical barrier difference between lithiation and sodiation to promote selectivity. The thermodynamic preference of $FePO_4$ provides baseline of selectivity while the particle features are critical to induce different kinetic pathways and barriers, resulting in different Li to Na selectivity from $6.2 \times 10^2$ to $2.3 \times 10^4$. Importantly, we categorize the $FePO_4$ particles into two groups based on their distinctly paired phase evolutions upon lithiation and sodiation, and generate quantitative correlation maps among Li preference, morphological features, and electrochemical properties. By selecting $FePO_4$ particles with specific features, we demonstrate fast (636 mA/g) Li extraction from a high Li source (1: 100 Li to Na) with $(96.6 \pm 0.2)\%$ purity, and high selectivity $(2.3 \times 10^4)$ from a low Li source (1: 1000 Li to Na) with $(95.8 \pm 0.3)\%$ purity in a single step.

Electrochemical intercalation has emerged as a promising method for selective Li extraction from unconventional sources to mitigate the Li supply issue[1–11]. One-dimensional (1D) olivine iron phosphate ($FePO_4$) has drawn tremendous attention due to its thermodynamic $Li^+$ intercalation preference, low $Li^+$ migration barrier, appropriate operating potentials within the water safety window, robust polyanionic structure, and demonstrated Li extraction selectivity and stability in authentic and simulated unconventional water sources (e.g., seawater)[1,3,4,7,8,12–14]. However, even with the intrinsic structural Li preference, during electrochemical extraction at low $Li^+$ concentrations or

atomic ratios, co-intercalation of interfering ions may occur, especially for the dominant competitor $Na^+$ ions (e.g., at a molar ratio of 1:1000 and below)[8]. Additionally, despite the widely recognized promise for Li extraction, the reported Li selectivity values range by nearly three orders of magnitude when using the same olivine-type $FePO_4$[1,3,4,7,15,16]. Besides the applied electrochemical methods, such discrepancies could be mainly due to the particle attributes adopted.

The selectivity of Li to Na in olivine $FePO_4$ is determined by both the thermodynamic preference and kinetic pathways during co-intercalation[7]. As a model material with anisotropic phase

[1]Pritzker School of Molecular Engineering, University of Chicago, Chicago, IL 60637, USA. [2]Department of Mechanical, Materials and Aerospace Engineering, Illinois Institute of Technology, Chicago, IL 60616, USA. [3]Center for Advanced Radiation Sources, University of Chicago, Chicago, IL 60637, USA. [4]James Frank Institute, University of Chicago, Chicago, IL 60637, USA. [5]NSF's ChemMatCARS, Pritzker School of Molecular Engineering, University of Chicago, Chicago, IL 60637, USA. [6]X-Ray Science Division, Advanced Photon Source, Argonne National Laboratory, Lemont, IL 60439, USA. [7]Department of Materials Design and Innovation, University at Buffalo, The State University of New York, Buffalo, NY 14260, USA. ✉e-mail: chongliu@uchicago.edu

transformation and preferred 1D migration along [010] direction[12,13,17–19], the intercalation behaviors of olivine $FePO_4$ depend highly on the morphology and size of the particle. Substantial differences exist between single-component $Li^+$ and $Na^+$ intercalation. As illustrated in Fig. 1, during lithiation, when the particle size reaches the critical nano-size region, both the nucleation barrier and miscibility gap vanish[20,21]. Single-phase solid solution (SS) transition occurs due to fast diffusion and elastically unfavorable phase separation[22–24]. For micron-sized $Li_xFePO_4$ ($0 < x < 1$), at low (de)lithiation rates, phase separation via spinodal decomposition dominates the transition[25,26]. Crystal anisotropy can lead to striped phase patterns in equilibrium which affects the spatial distribution of Li[22]. It is worth mentioning that, for micron-sized particles, quasi-solid solutions could be realized under large currents, during which the (de)lithiation time is too short for complete phase separation[27–31]. Moreover, a further increase in the particle size raises the coherency strain energy. This leads to phase transition out of mechanical equilibrium, which rarely occurs due to small volume change for (de)lithiation and was only observed at higher rates[20,27]. In contrast, the phase transition is different during sodiation. Mechanical nonequilibrium can be easily induced due to the large volume expansion upon transition to $NaFePO_4$ (16.6%)[32]. An intermediate buffer phase, $Na_{2/3}FePO_4$, is needed to mitigate the volume expansion even for small particles (under the structural equilibrium). At slow (de)sodiation rates, olivine $Na_yFePO_4$ will separate into $FePO_4$ and $Na_{2/3}FePO_4$ phases for $0 < y < 2/3$ and remain in a solid-solution transition for $2/3 < y < 1$[32]. Equilibrium solid solution transition throughout the range ($0 < y < 1$) during sodiation has not been seen experimentally, even though the particle size reaches the critical size for lithiation. Moreover, morphology also plays critical roles in determining the intercalation pathway. Platelet particles with a preferred (010) facet have a much lower exchange current than ellipsoidal particles with similar size, which could increase the active particle population and promote uniform SS domains under the same applied current[28,33]. Further, when transforming from $FePO_4$ to $Li_xFePO_4$ or $Na_yFePO_4$, the stable interface orientation is shown to depend both on the particle size and morphology, due to different anisotropies, interfacial energy, and coherency strain penalty[34]. Despite the rich knowledge of lithiation and sodiation, the effect of morphology and size on the co-intercalation behavior in $FePO_4$ and Li selectivity is largely unknown and unpredictable, which limits the rational design of host materials for Li extraction.

To reveal the key features of $FePO_4$ particles that govern the Li selectivity, we designed and synthesized a series of particles with different morphologies and host responses upon $Li^+$-$Na^+$ co-intercalation. We unveil that, to realize high Li selectivity in extremely dilute sources, it is critical for the $FePO_4$ particle to reach a threshold dimension where the kinetic barrier for sodiation can be harnessed to enlarge the energy differences between $Li^+$ and $Na^+$ intercalation. This threshold dimension delineates phase transformation behaviors into two distinct groups, as illustrated in Fig. 1. One group exhibits equilibrium solid solution lithiation transition paired with equilibrium phase separation sodiation transition, while the other group displays equilibrium phase separation lithiation transition alongside an out-of-equilibrium sodiation transition. The first group comprises small particles with channel lengths less than 100 nm, while the second group consists of particles with channel lengths exceeding 500 nm. In situ synchrotron X-ray diffraction revealed that the larger particle group experienced pronounced lattice distortion during sodiation instead of a responsive phase transition due to the significantly increased nucleation barrier and coherency strain energy, which builds up large overpotential in

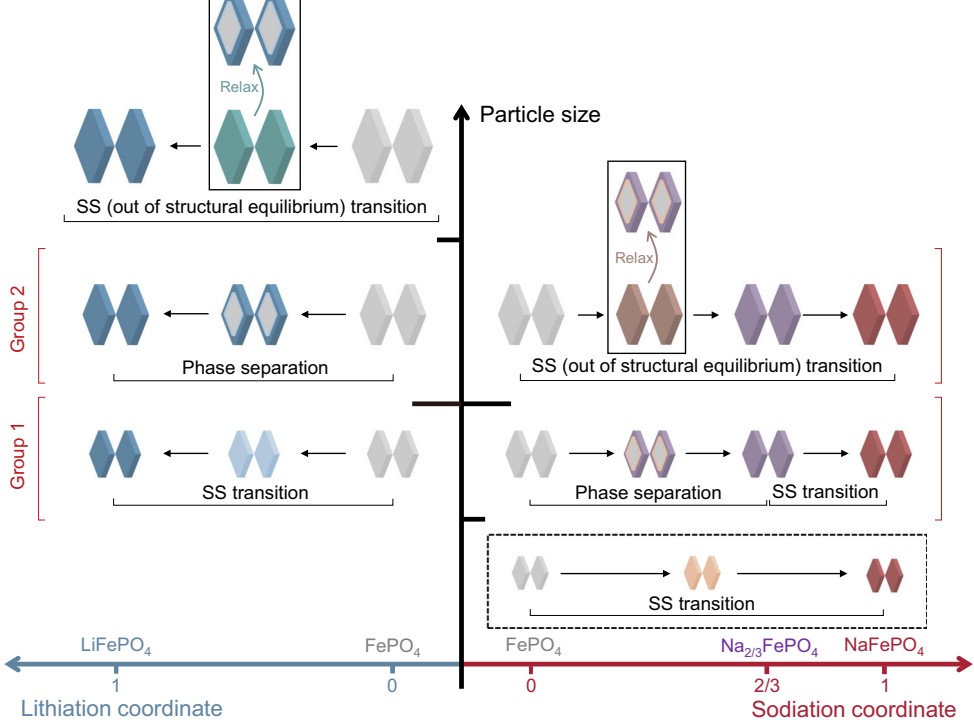

**Fig. 1 | Schematic illustrations depicting the particle size dependent phase evolutions of olivine $FePO_4$ particles during lithiation or sodiation.** Different color codes denote different phases during lithiation or sodiation. Here, we grouped the particles based on their different phase evolution pathways upon lithiation and sodiation. Some previous works also witnessed some phase transformations, including solid solution (SS) transition during lithiation[20–22,28], phase separation transition during lithiation[25,26], SS transition out of structural equilibrium during lithiation[27–31], and two-stage sodiation transition (phase separation + SS transition)[32]. In this work, we observed SS (out of structural equilibrium) transition upon sodiation. The dashed box in the diagram indicates the equilibrium SS transition throughout the range upon sodiation has not been observed experimentally.

kinetics and mechanics for sodiation, increasing Li preference. For small particles, the selectivity primarily arises from thermodynamic Li preference, and kinetic Li preference can only be induced at extremely high currents. Due to the minimal nucleation barrier and rapid diffusion for $Na^+$, the kinetic preference for Li will quickly diminish via considerable non-faradaic ion exchange between electrolyte $Na^+$ and structural $Li^+$. Quantitative correlation maps linking Li extraction performance to particle electrochemical properties and morphological features revealed strong correlations between 1) $FePO_4$ electrochemical characteristics (e.g., kinetic barrier difference) and Li selectivity, and 2) particle features (e.g., [010] channel length and particle volume) and Li selectivity. The correlations indicate the existence of an optimal size range ([010] length 155–420 nm) for achieving both high Li selectivity and structural reversibility. Guided by our discovery, by choosing $FePO_4$ particles with different features, in a single step, we achieved fast (636 mA/g) Li extraction from high Li source (1: 100 Li to Na) with $(96.6 \pm 0.2)\%$ purity, and high selectivity $(2.3 \times 10^4)$ Li extraction from low Li source (1: 1000 Li to Na) with $(95.8 \pm 0.3)\%$ purity.

## Results

### Quantification of particle morphology features and electrochemical response during lithiation or sodiation

As illustrated in Fig. 2a, when a negative potential is applied, Li/Na ions in the electrolyte first accumulate on the (010) channel openings before leaping across the carbon coating into the interstitial vacancies present in the first layer of the crystal, while the electrons in the carbon coating tunnel to the adjacent iron site to reduce the $Fe^{3+}$ ions. After the charge transfer reaction, adjacent $Fe^{2+}$ and Li/Na ions form a neutral quasiparticle, or polaron, capable of migrating along the preferred [010] channels[35]. Notably, depending on the particle features and electrochemical response upon lithiation or sodiation, phase evolutions can manifest as two-phase separation or solid solution transition. It becomes evident that controlling particle morphology, encompassing the (010) facet, [010] channels, as well as particle volumes, is of utmost importance, given the anisotropic nature of ion transport in olivine $FePO_4$ crystal. Specifically, the relative area of each facet of a particle depends on its surface energy[36]. According to our constructed Wulff shape of $LiFePO_4$ from calculated surface energies (Fig. 2b, Supplementary Table 1, and Supplementary Note 1 for computation details), surfaces (201), (100), and (010) have the lowest energies, which are also consistent with reported results[36]. Here, six distinct well-crystallized $LiFePO_4$ particles were prepared using solvothermal approaches followed by the surface carbon-coating treatment under calcination (Fig. 2c-h, Supplementary Figs. 1-6, and See Methods and Supplementary Note 2 for more synthesis details). Based on Rietveld refinement, the anti-site defect level is estimated to be low for all six particles (Supplementary Fig. 7 and Supplementary Table 2). The delivered capacities, which will be discussed later, further verify the low defect concentrations, such as less than 0.1% for the biggest particle (Cuboid-6000 nm). The facets of synthesized particles predominantly exhibit two orientations, either (010)-oriented (platelet particles) or (100)-oriented (cuboid particles), both of which have low surface energies (Fig. 2b, Supplementary Table 1). Specifically, one of the most crucial morphology features, the [010] channel length (1D migration direction) covers a wide range with average dimensions of 20, 45, 87, 600, 1200, and 6000 nm, respectively (Fig. 2c-h and Supplementary Figs. 1-6). More morphology features are considered to provide a comprehensive quantification of the size and morphology. As shown in Fig. 2i, Supplementary Table 3, and Supplementary Note 3, the average particle length in the [100] and [001] directions are also quantified. Additionally, we determine the average exposure ratio of the (010) facet to the total surface area, a metric ranging from 12% to 70%. Furthermore, the (010) facet area to [010] channel length ratio is evaluated (ranging from $2.37 \times 10^2$ nm to $1.67 \times 10^4$ nm), which reflects the accessibility of storage sites and can influence the exchange

current density of the particles[28,33,34,37]. We also estimate the average particle volume by calculating the product of the (010) area and the [010] channel length, which ranges between $2.5 \times 10^{-4}$ $\mu m^3$ and 24 $\mu m^3$.

The electrochemical lithiation and sodiation behaviors of each particle were characterized next. The empty $FePO_4$ hosts were prepared by chemical Li extraction (See Methods for chemical extraction and electrode preparation details), with the structure verification from Rietveld refinement (Supplementary Fig. 8 and Supplementary Table 4). We first compared the constant current (de)intercalation curves of each particle in 1 M LiCl or 1 M NaCl aqueous solutions (Fig. 3a-d, Supplementary Figs. 9-10, and Supplementary Table 5). The decent capacity delivered during delithiation at 0.1 C (17 mA/g), ranged from 129 mAh/g for Cuboid-6000 nm particles to 159 mAh/g for Platelet-20 nm and Platelet-600 nm particles, indicating the low Li-Fe anti-site defects level, especially when considering the channel length (e.g., <0.1% for Cuboid-6000 nm particles)[24].

The intercalation voltage difference between $Li^+$ and $Na^+$ is a good indicator for Li selectivity. The six particles exhibited distinctive group behaviors based on their lithiation C-rate response and sodiation behavior, leading us to categorize them into two groups. As shown in Fig. 3a-d, Supplementary Figs. 9-10, one group, which consists of small particles with channel length <100 nm (Group 1), displayed minimal (de)lithiation voltage hysteresis and smaller hysteresis during (de)sodiation than the other group. Particles within this group demonstrated excellent rate capability during (de)lithiation at 0.5 C as well, suggesting faster kinetics. Conversely, notable differences in voltage hysteresis emerged during (de)sodiation for Platelet-600 nm, Platelet-1200 nm, and Cuboid-6000 nm particles, leading to their categorization into a separate group (Group 2). Furthermore, all three particles in the larger particle group experienced a reduced capacity retention during their first desodiation (Supplementary Table 5). Particularly noteworthy is the potential difference at the halfway capacity point of the initial sodiation at 0.1 C, which can reach up to 0.36 V between the two groups (Supplementary Table 6). This difference can be attributed to the more pronounced strain/nucleation penalty and slower kinetics experienced by the larger particle group. Additionally, the big particles demonstrate a high degree of non-topochemical $Na^+$ intercalation, as indicated by the decreased capacity retention at the first charge (Supplementary Table 5). It is worth highlighting that the Platelet-600 nm particle in Group 2 has the largest difference in cycling features between (de)lithiation and (de)sodiation. Platelet-600 nm demonstrate better (de)lithiation rate capability; however, both Platelet-1200 nm and Cuboid-6000 nm particles displayed significant capacity decay during (de)lithiation at 0.5 C (Supplementary Fig. 10). In summary, the size of the Platelet-600 nm particle is relatively small to release strain penalty and facilitate fast $Li^+$ (de)intercalation kinetics but is large enough to induce kinetic and chemomechanical barriers during $Na^+$ (de)intercalation.

To better isolate the overpotential gain attributed to kinetics and mechanics from thermodynamic energy difference, we monitored the potential change throughout the constant current intercalation until reaching a certain depth of discharge, followed by 20 h of relaxation in the original solution (Fig. 3e and Supplementary Fig. 11). The voltage difference was labeled in bars with bottom and top boundaries indicating voltages after intercalation and relaxation, respectively. Preconditioned $FePO_4$ particles (Cycled once at 17 mAh/g in 1 M LiCl aqueous solution to extract the accessible capacity delivered at the first charge) are used here to follow the steps for the later $Li^+$-$Na^+$ co-intercalation process. Specifically, the calculations of applied current and depth of lithiation or sodiation are based on the delivered capacity in the first de-lithiation rather than the theoretical capacity. For instance, 0.1 C' for the Platelet-20 nm particle corresponds to 15.9 mA/g (Supplementary Table 5), and DOD_Li0.35'/Na0.35' represents 55.65 mAh/g capacity usage. As depicted in the left panel of Fig. 3e and

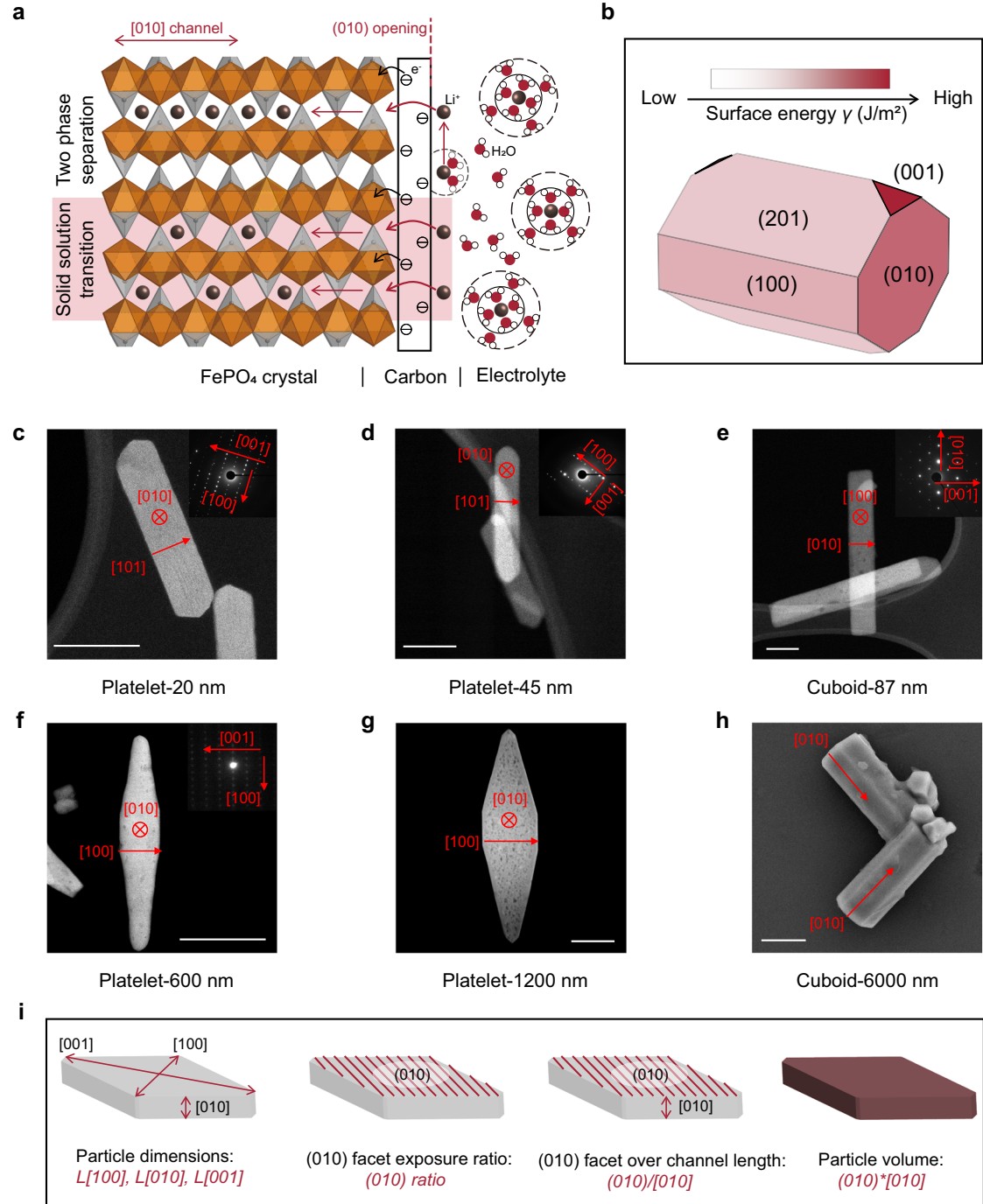

**Fig. 2 | Particle morphology features. a** Schematic illustration of the ion insertion process within a carbon-coated FePO$_4$ crystal. Ion enters from the (010) facet and migrates along the [010] direction. **b** Constructed Wulff shape of LiFePO$_4$ using the calculated surface energies of specific orientations. **c–g** Scanning transmission electron micrographs (STEM) and selected area electron diffraction (SAED) patterns (top-right) of Platelet-20 nm (**c**), Platelet-45 nm (**d**), Cuboid-87 nm (**e**),

Platelet-600 nm (**f**) and Platelet-1200 nm (**g**) particles. The SAED patterns were taken along the axis labeled in the red across. The red arrows denote some specific orientations of the particles. Scale bars in **c-e**, 100 nm. Scale bars in **f-g**, 2 μm. **h** Scanning electron micrograph (SEM) of Cuboid-6000 nm particles. Scale bars, 2 μm. **i** Schematic diagram illustrating some characteristic morphology features.

summarized in Supplementary Table 7, all particles, except Platelet-1200 nm and Cuboid-6000 nm, have relatively small lithiation barriers. The operando lithiation potential and the equilibrium potential after relaxation both have small differences and are close to the thermodynamic voltage of Li$_x$FePO$_4$ (0.259 V vs. Ag/AgCl; See Supplementary Note 1 for computation details), which indicates a low kinetic/chemo-mechanical lithiation barrier at 0.1 C'. For larger particles (channel length > 1000 nm), kinetic barriers scale significantly with size. For

example, at DOD_Li70', the end intercalation potential can be ~ 0.1 V lower for Platelet-1200 nm and Cuboid-6000 nm particles. During 0.1 C' sodiation, as shown in the right panel of Fig. 3e, all particles exhibited higher overpotential relative to the thermodynamic sodiation voltage. This deviation can be attributed to the sluggish intercalation of Na$^+$ ions, accompanied by increased nucleation or strain energy penalties. Additionally, the Group 2 particles exhibited larger overpotentials (summarized in Supplementary Table 7), further

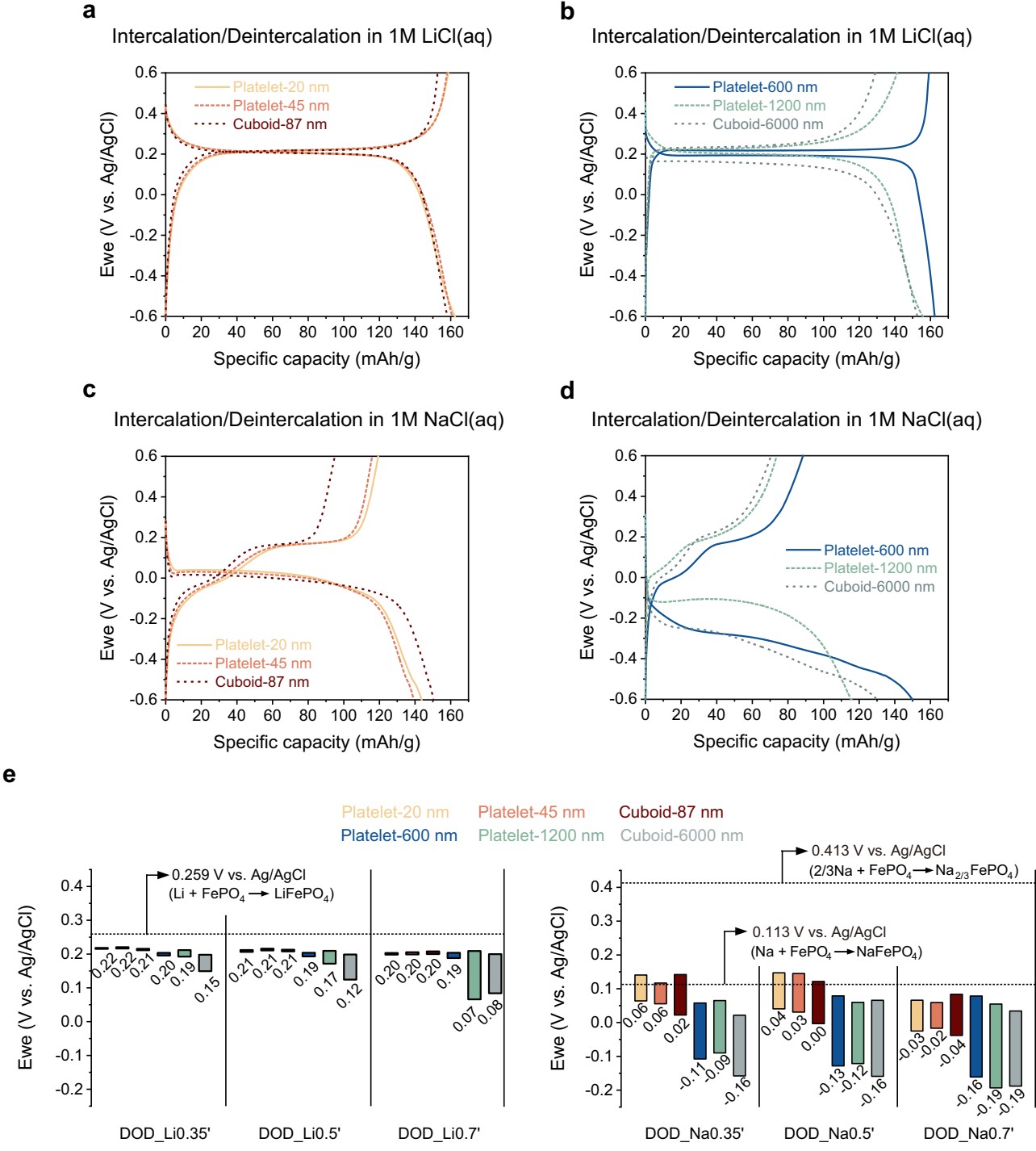

**Fig. 3 | Electrochemical response during lithiation or sodiation. a, b** 1st electrochemical cycle under 17 mA/g (equivalent to 0.1 C based on theoretical capacity of LiFePO₄) in 60 ml 1 M LiCl aqueous solution (paired with Ag/AgCl/KCl (4.0 M) reference and Li$_x$FePO₄ counter electrodes). **c, d** 1st electrochemical cycle under 15.4 mA/g (equivalent to 0.1 C based on theoretical capacity of NaFePO₄) in 60 ml 1 M NaCl aqueous solution (paired with Ag/AgCl/KCl (4.0 M) reference and Na$_y$FePO₄ counter electrodes). **e** Bar chart comparisons of end potential collected right after different depth-of-discharge (DOD) in 60 ml 1 M LiCl (left panel) or NaCl (right panel) aqueous solution, which corresponds to the value at the bottom of the bar, and open-circuit potential after 20 h of relaxation without currents, corresponding to the value at the top of the bar. See Methods for electrode preparation and DOD calculation. The dashed lines denotes the calculated thermodynamic voltage for specific reactions (See Supplementary Note 1 for computation details).

highlighting the more pronounced effects of kinetics and mechanics on big particles.

This rich collection of particle morphology characteristics and electrochemical responses will be used to identify critical features associated with high Li preference.

## Particle morphology-dependent phase evolutions during lithiation or sodiation

The intercalation pathways and the associated phase evolutions or ion storage mechanisms are also critical in determining the energy barriers for both Li⁺ and Na⁺ intercalation, consequently influencing Li

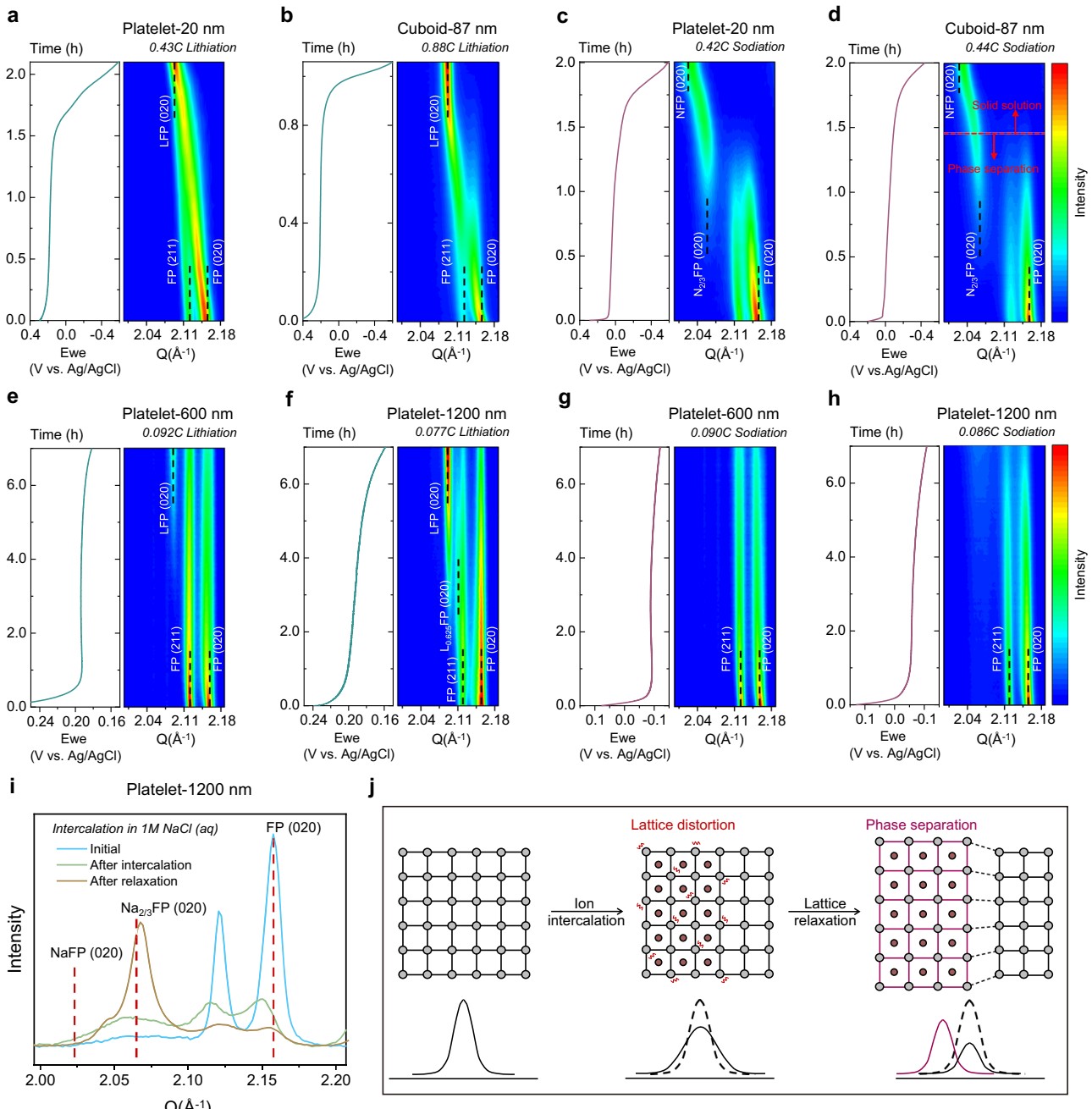

**Fig. 4 | In situ synchrotron X-ray diffraction (XRD) tracking of phase evolutions during lithiation or sodiation. a** Lithiation of Platelet-20 nm particles at 0.43 C. **b** Lithiation of Cuboid-87 nm particles at 0.88 C. **c** Sodiation of Platelet-20 nm particles at 0.42 C. **d** Sodiation of Cuboid-87 nm particles at 0.44 C. **e** Lithiation of Platelet-600 nm particles at 0.092 C. **f** Lithiation of Platelet-1200 nm particles at 0.077 C. **g** Sodiation of Platelet-600 nm particles at 0.090 C. **h** Sodiation of

Platelet-1200 nm particles at 0.086 C. **i**, Snapshots (initial scan and last scan) of in situ synchrotron XRD during sodiation as well as the ex situ synchrotron XRD of the electrodes after ~ 10 h relaxation in the open air for Platelet-1200 nm particles. **j**, Schematic showing the lattice distortion and relaxation processes with the corresponding XRD peak features.

preference[7]. To investigate the host response upon lithiation or sodiation, we used in situ synchrotron X-ray diffraction (XRD) to track the structure changes. Figure 4a-b, and Supplementary Fig. 12 reveal that, during lithiation, particles in Group 1 (channel length <100 nm) undergo SS transitions from the initial FePO$_4$ phase, evidenced by a continuous change of peak positions and lattice parameters. These solid solution phases are thermodynamically stable and persist even at lower lithiation rates, such as 0.1 C or 0.01 C (Supplementary Fig. 13). The SS crystallographic insertion pathway indicates the vanishing of the miscibility gap and improved kinetics, which have been seen by previous works[21,38] and are also consistent with our observed small

voltage hysteresis (Fig. 3a, and Supplementary Figs. 9-10). For Group 1 particles, during sodiation, a two-stage phase evolution pathway was observed (Fig. 4c-d, and Supplementary Fig. 14). This pathway follows the phase diagram of Na$_y$FePO$_4$ (0 < y < 1) proposed by Lu et al.[32], commencing with a two-phase equilibrium between FePO$_4$ and Na$_{2/3}$FePO$_4$ phases, then progressing into an SS transition from Na$_{2/3}$FePO$_4$ to NaFePO$_4$ phase. It is worth noting that, at the first stage of sodiation, the smallest Platelet-20 nm particle shows broader peaks and more intensity contributions from the intermediate compositions compared with the Cuboid-87 nm particle, which suggests some degree of solid solution formation, but the transition is still dominated by the two-

phase equilibrium. Interestingly, even when doubling the sodiation rate for the Cuboid-87 nm particle, we couldn't realize the out-of-equilibrium SS transition between $FePO_4$ and $Na_{2/3}FePO_4$ phases (Supplementary Fig. 14b). This underscores the benefit of the intermediate $Na_{2/3}FePO_4$ phase formation in mitigating the volumetric strain during sodiation.

The relatively worse rate capability of Group 2 particles requires us to employ a slower lithiation or sodiation rate. During lithiation, phase-separation-dominated evolutions are witnessed for all three big particles (Fig. 4e, f, Supplementary Figs. 15 and 16a-b), which is consistent with previous studies[17,25,39]. Specifically, in the case of Platelet-600 nm particles, there is a noticeable broadening and a slight left shift of the $FePO_4$ (020) peak in the beginning of lithiation, followed by the co-existence of $LiFePO_4$ and $FePO_4$ phases. This is due to a certain level of intrinsic Li solubility. Interestingly, for the Platelet-1200 nm particle, additional features become apparent. (200) and (020) peaks originating from the intermediate phase $Li_{0.625}FePO_4$ manifest before the formation of the $LiFePO_4$ phase (Fig. 4f and Supplementary Fig. 16b). $Li_{0.625}FePO_4$ is situated at the eutectoid point of the phase diagram and has been observed as a preferred intermediate phase at high currents[39]. Notably, it is intriguing that this intermediate phase can persist even at a relatively slow current (0.077 C), and we attribute this to the higher (020) facet exposure ratio and more accessible storage sites ((020) facet area/[020] channel length). The presence of the $Li_{0.625}FePO_4$ eutectoid composition will assist in releasing volumetric strain and elevate the lithiation voltage shown in Fig. 3, analogous to the $Na_{2/3}FePO_4$ buffer phase. Furthermore, the emergence of the intermediate composition is consistent with the more slanted chronopotentiometry curve observed compared to the Platelet-600 nm particle (Fig. 4e, f). Additional unexpected features were observed for the larger particles during sodiation (e.g., Platelet-600 nm and Platelet-1200 nm particles). As illustrated in Fig. 4g-i, and the snapshots in Supplementary Fig. 17, the sodiation process primarily involves peak intensity decrease, notable peak broadening, and a slight left shift, without distinct phase transformations. Moreover, the peaks exhibit reduced symmetry during in situ sodiation which indicates strong lattice distortions[27]. Interestingly, when the electrodes were allowed to relax in the open air overnight (~10 h), subsequent ex situ synchrotron XRD revealed the presence of $Na_{2/3}FePO_4$, which represents the thermodynamic equilibrium phase. More specifically, compared to Platelet-600 nm particles, Platelet-1200 nm particles already started to transit to $Na_{2/3}FePO_4$ phase at the late stage of in situ sodiation before relaxation (Supplementary Fig. 16d), which demonstrates better capability to release the chemo-mechanical strain and also consistent with the observed higher sodiation voltage (Fig. 3d). Overall, these observations suggest significant lattice distortion occurring during $Na^+$ intercalation (Fig. 4j). The pronounced volumetric strain and nucleation energy penalty experienced by the big particles disrupt their structural equilibrium, thereby suppressing in situ phase separation. The phase response of the larger particles is also consistent with the previously observed high overpotential and considerable kinetic barriers (Fig. 3). The observed phase evolutions confirm the rationale behind the grouping of particles based on their morphological form factors that particles in Group 1 have SS lithiation evolution pathway (in equilibrium) paired with two-stage phase evolution pathway during sodiation (in equilibrium), while particles in Group 2 have phase-separation-dominated lithiation evolutions (in equilibrium) paired with out-of-equilibrium sodiation transition.

## Li extraction performance and non-faradaic ion-exchange in 1D $Li_xFePO_4$ hosts

The Li extraction performance of the six particles was examined using 1 mM: 1 M Li to Na molar ratio solutions unless specified. As shown in Fig. 5a, employing 70% accessible capacity and a 0.1 C' extraction rate, particles in Group 2 exhibited better Li selectivity than those in Group

1. Particularly, the Platelet-600 nm particle showed the highest Li preference (recovered Li/(Li+Na) ratio = 0.95 ± 0.012) with a Li selectivity of $2.1 \times 10^4$, approximately 34-fold higher than that of the Platelet-20 nm particle. We further evaluated the effects of co-intercalation rates in Fig. 5b. Platelet-600 nm particles showed a monotonic decrease in Li selectivity with elevated extraction rates. As shown in Supplementary Fig. 18, when applying a small current (≤0.5 C'), the overpotential is small so that there is not enough energy to overcome the kinetic, chemo-mechanical, and nucleation barriers associated with sodiation, resulting in excellent Li selectivity ranging from $8.5 \times 10^3$ to $2.1 \times 10^4$. However, as the overpotential increases at higher currents (>0.5 C'), mass transfer limitations of $Li^+$ ion become more pronounced. The larger overpotentials overcome the energy barriers for $Na^+$ intercalation, leading to a decline in selectivity (<$9.6 \times 10^2$). The substantial overpotential eventually leads to less than 70% accessible capacity at voltage cutoff.

Interestingly, for the small particles in Group 1, we observed a non-monotonic trend in Li selectivity as the extraction rates increased. The drop in selectivity at a high extraction rate (e.g., 1.0 C'/2.0 C') was expected due to mass transfer limitations on the electrolyte side caused by low $Li^+$ concentrations (1 mM), similar to what was observed in the larger particles. At lower rates below 0.5 C', the differences between $Li^+$ and $Na^+$ intercalation enlarge. This is substantiated by examining the sodiation and lithiation chronopotentiometry curves obtained in pure 1 M NaCl(aq) and 1 M LiCl(aq) under varying rates (Supplementary Fig. 19). At 50% depth-of-lithiation/sodiation, more considerable sodiation barriers (~0.05 V) were witnessed from 0.1 C' to 1.0 C', while there was almost no increase in end potential at elevated lithiation rates, which can be attributed to the excellent rate capability of the small particles. Therefore, the increase in Li selectivity is attributed to the kinetic barrier gained from sluggish $Na^+$ intercalation.

To take advantage of the rate capability of small particles, we can intentionally increase the kinetic barrier of $Na^+$ intercalation by using super-fast extraction rates (e.g., 6 C'). At elevated extraction rates, only brines with a higher $Li^+$ concentration are applicable (e.g., 10 mM: 1 M Li: Na) to circumvent the mass transfer limitations on the electrolyte side. In Fig. 5c, d, we conducted a comparison of the extraction performance between Platelet-20 nm and Platelet-600 nm particles at high extraction rates in a 10 mM: 1 M Li: Na (1:100) mixed solution. Platelet-20 nm and Platelet-600 nm particles were previously identified as the worst and best performers in Fig. 5a, at a low extraction rate (0.1 C') in a 1 mM: 1 M Li: Na (1:1000) mixed solution. Obviously, Platelet-20 nm particles outperformed Platelet-600 nm particles in all aspects at faster extraction rates (>4 C'). Lower energy input from the smaller overpotentials during extraction, as well as the higher Li selectivity, make small particles a better choice for brines with relatively high $Li^+$ concentrations, such as the biggest Li brine source, Atacama, in Chile (0.22 M: 4 M: 0.4 M Li: Na: Mg)[8]. In other words, for small particles, the selectivity primarily arises from thermodynamic Li preference. Due to the minimal nucleation barrier and rapid diffusion for both ions, the kinetic preference for Li during co-intercalation will quickly diminish when we use low extraction rates. Specifically, our calculated lithiation voltage of the olivine $FePO_4$ host is 0.259 V vs. Ag/AgCl, which is only 0.146 V higher than the sodiation voltage (0.113 V vs. Ag/AgCl) (See Supplementary Note 1 for more calculation details). The 0.146 V difference cannot bear the three orders of concentration difference between $Li^+$ (1 mM) and $Na^+$ (1 M) based on the Nernst equation, if there is no kinetic barrier gain.

Moreover, non-faradaic ion exchange experiments further verify the thermodynamic-dominated Li preference of small particles. Initially, we pre-intercalated pure $Li^+$ into the hosts at 0.1 C' until different depth-of-discharge (DOD_Li0.1'/0.35'/0.5'). Subsequently, we soaked the electrodes in a mixed solution containing 1 mM LiCl and 1 M NaCl while concurrently measuring the open circuit voltage (OCV) (See Methods for more details). Supplementary Fig. 20 shows significantly

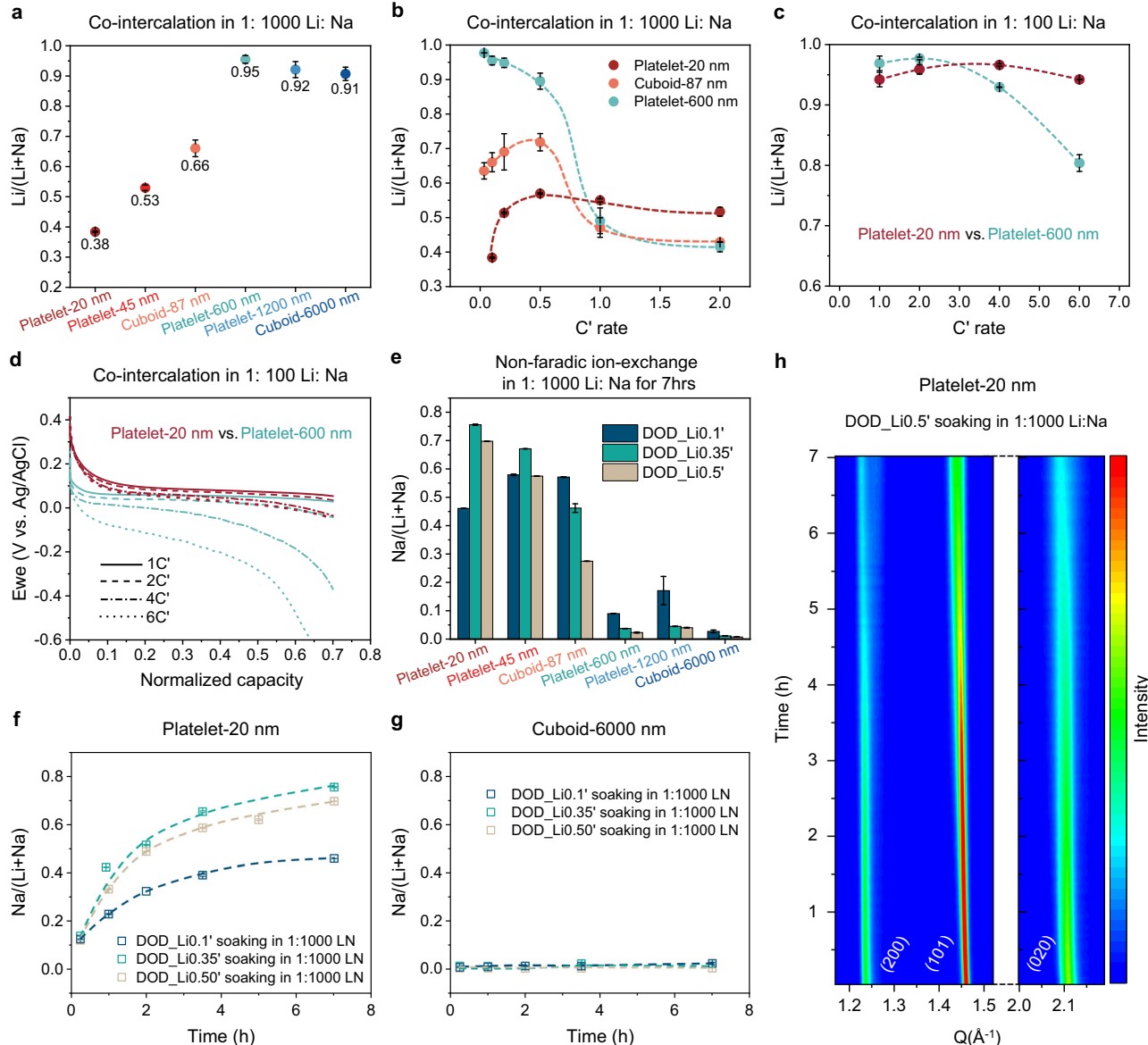

**Fig. 5 | Li extraction performance and non-faradaic ion-exchange. a** Li/(Li+Na) ratios after recovery of different electrodes from 1:1000 Li to Na solution using 70% accessible capacity and 0.1 C' extraction rate. **b, c** Li/(Li+Na) ratios after recovery of electrodes used under different extraction rates from 1:1000 Li to Na solution (**b**) and 1:100 Li to Na solution (**c**) using 70% accessible capacity. **d**, Intercalation curves of Platelet-20 nm and Platelet-600 nm particles under different extraction rates. **e** Measured Na/(Na+Li) ratios of the electrodes after soaking in 1 mM LiCl and 1 M NaCl mixed solution for 7 h, using Li pre-intercalated particles. DOD_Li0.1'/0.35'/ 0.5' denotes the seeding percentage based on the accessible capacity. **f, g** Measured Na/(Na+Li) ratios at different times of Platelet-20 nm and Cuboid-6000 nm particles after soaking in 1 mM LiCl and 1 M NaCl mixed solution, using DOD_Li0.1'/ 0.35'/0.5' pre-intercalated particles. **h** In situ synchrotron XRD tracking of Platelet-20 nm pre-intercalated particles (DOD_Li0.50') during ion-exchange in 1 mM LiCl and 1 M NaCl mixed solution. Mass loading for electrodes used in **a–g** is ~ 2.5 mg/ cm². Error bars represent the standard deviation of three replicate measurements.

different OCV curves between the two particle groups, although they all reached equilibrium after around 7 h. For big particles, an exponential decay of the potential was observed and stabilized within one hour. In contrast, the small particles exhibited a peculiar upward tilt of the curve after the initial decay, bringing it closer to the equilibrium voltage in pure Na solutions. We then measured the Na/Li contents in the particles. As shown in Fig. 5e, substantial non-faradaic ion exchange was witnessed for the Group 1 small particles at all depths of pre-lithiation. Specifically, in the case of DOD_Li0.35', 75.6 ± 0.2% of the structure Li⁺ in Platelet-20 nm particles was replaced by the solution Na⁺, whereas less than 5% exchange was observed for the three big particles. We also monitored the composition evolutions overtime during the soaking process (Fig. 5f, g and Supplementary Fig. 21). The non-faradaic ion exchange behavior indicates that the kinetics of Li⁺

intercalation is faster than Na⁺ intercalation at certain C rate ranges; however, since the thermodynamic preference for Li alone cannot tolerate three-order-of-magnitude difference in Li⁺ and Na⁺ concentrations, a significant amount of Na⁺ will slowly replace structural Li⁺ via ion exchange. The much smaller nucleation barrier and more rapid ion diffusion in Group 1 small particles facilitate such a significant non-faradaic ion exchange process within the 1D olivine FePO₄ hosts, which has not been reported before. Consequently, at higher C rates, the shorter amount of contact time can also reduce the degree of ion exchange and promote Li selectivity. The phase evolution during ion exchange was also tracked by the in situ synchrotron XRD (Fig. 5h and Supplementary Fig. 22). Soaking Platelet-20 nm DOD_Li0.5' particles in 1 mM: 1 M Li: Na mixed solution, we observed a continuous broadening of the peaks and a decrease in intensity, particularly in the in-plane

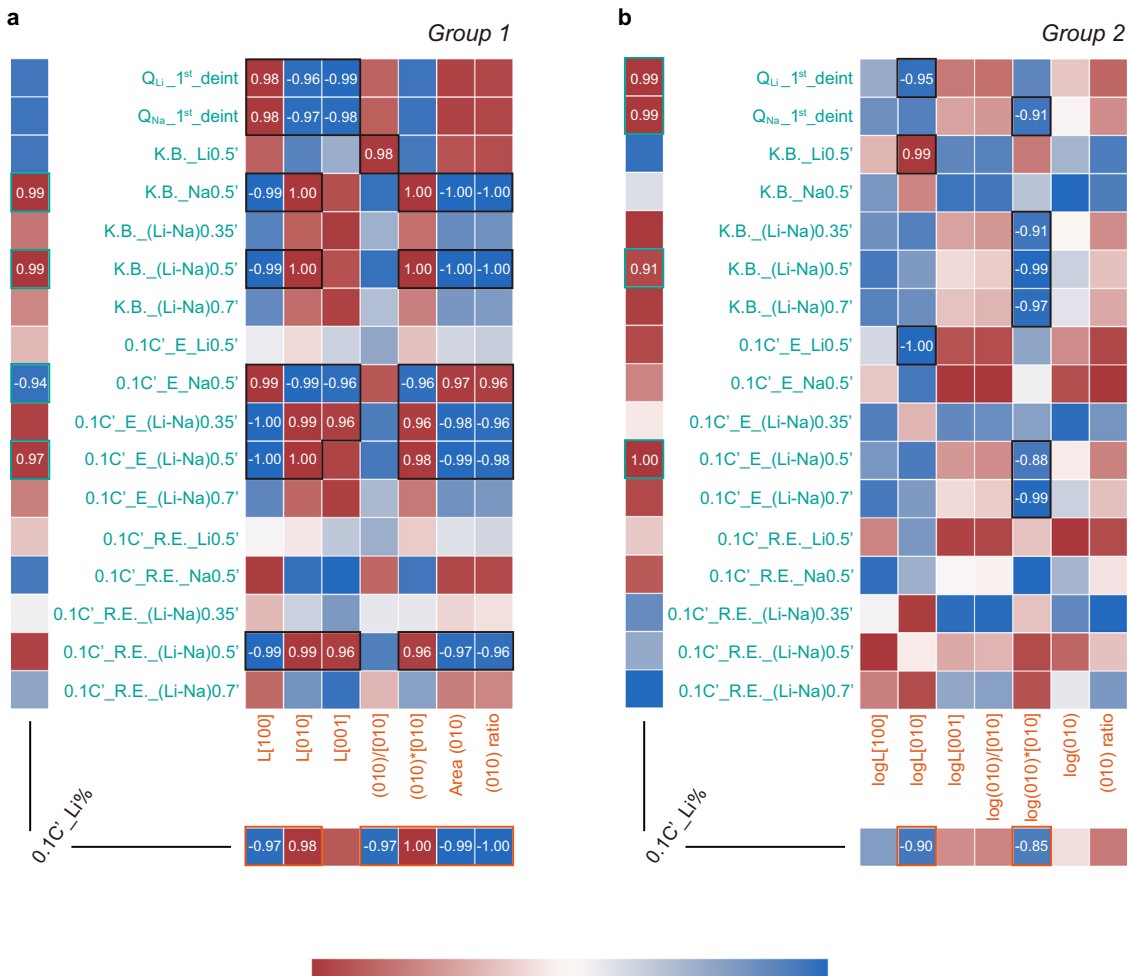

**Fig. 6 | Quantitative correlation maps for each group illustrating Li competitiveness, particle morphology, and electrochemical characteristics features.** **a** Coefficient of correlation (R) map for particles in Group 1. **b** Coefficient of correlation (R) map for particles in Group 2. Each value calculated in this map represents the degree of relationship between two variables. Specifically, "*0.1 C' Li%*" represents the recovered Li/(Li+Na) atomic ratios from 1 mM:1 M LiCl: NaCl(aq) mixed solutions at 0.1 C'. Features labeled in orange correspond to particle morphology features, while those labeled in green pertain to electrochemical characteristics features. Specifically, "$Q_{Li/Na}$" denotes the delivered capacity in the first de-lithiation/sodiation process; "*K.B.*" denotes the measured kinetic barrier/potential change during relaxation after different depths of lithiation/sodiation (e.g., 35/50/70%); "*0.1 C' E*" denotes the end potential after 0.1 C' constant current lithiation/sodiation until different depths of discharge (e.g., 35/50/70%); "*R.E.*" denotes the end potential after relaxation; "*L*" denotes the length of the particle along specific directions (e.g., [100]/[010]/[001]); "*(010)/[010]*" denotes the (010) facet area over [010] channel length ratio; "*(010)\*[010]*" denotes the particle volume; "*Area (010)*" denotes the (010) facet exposure area; "*(010) ratio*" denotes the (010) facet exposure area over total surface area; features with "*log*" prefix denote the corresponding logarithmic values; See Supplementary Tables 8, 9, and Supplementary Note 3 for the complete definition of each variable or summary of the values for each particle.

direction, such as (101) facet. This indicates a more disordered structure and a shorter coherence length in the in-plane directions after ion exchange. Given the highly confined 1D structure of olivine FePO₄ hosts, the observed non-faradaic ion exchange process is both intriguing and unexpected. Further investigations are necessary to fully comprehend the underlying mechanism behind this phenomenon.

**Identification of critical features with high Li preference**

To identify the relationships of Li competitiveness to particle morphology and electrochemical characteristics, we generated correlation coefficient (R) maps for both groups of particles in Fig. 6, Supplementary Figs. 23-24. These maps quantitatively represent the extent of linear relationships between any two variables (See Supplementary Tables 8-9, Supplementary Note 3 for the definition of each variable and summary of the values for each particle). The R values fall within the range of -1 to 1. A correlation coefficient of 1 or -1 means a perfect positive or negative correlation, respectively.

Most importantly, the analysis identifies critical morphology and electrochemical features that indicates Li preference. For electrochemical characteristics, the operando voltage difference (e.g., *0.1 C' E_(Li-Na)0.5'*) and kinetic barrier difference (e.g., *K.B._(Li-Na)0.5'*) between Li and Na show a strong positive correlation with Li preference for both particle groups. However, the dependence of Li preference on resting voltage difference (e.g., *0.1 C' R.E._(Li-Na)0.5'*), which reflects the thermodynamic properties, is relatively weak for Group 2 particles. This indicates that the thermodynamic preference provides the baseline for Li selectivity but the morphologies of FePO₄ particles can be further designed to increase the kinetic barrier differences between Li and Na to promote the Li selectivity. For both Group 1 and Group 2 particles, kinetic barrier difference, *K.B._(Li-Na)0.5'*, and

operando intercalation voltage difference, $0.1 C'\_E\_(Li-Na)0.5'$, are identified as general descriptors for Li selectivity.

For morphology features, Group 1 small particles show very sensitive dimension dependence of Li preference that [010] channel length ($L[010]$) and (010) area ($Area\ (010)$) showed strong positive and negative correlation, respectively. Therefore, $(010)/[010]$, $(010)*[010]$, and $(010)\ ratio$ all show a strong correlation with Li selectivity. The negative correlation of $Area\ (010)$ could be due to the non-faradaic ion exchange behavior observed that a larger (010) area increases the reaction interface, allowing for a greater exchange of $Na^+$ ions into the $FePO_4$ host. For Group 2, strong correlations were identified for $log[010]$ and $log(010)*[010]$. The logarithmic positive growth pattern of these morphological features indicates that excessively large particles are unnecessary to enhance Li favorability. Besides, the correlation between $log(010)*[010]$ and kinetic barrier differences is even stronger than that of $log[010]$. However, $Area\ (010)$ displays a weak correlation. This underscores the significance of mechanical deformation in influencing Li preference, as demonstrated by the robust correlation of $(010)*[010]$ (particle volume). Moreover, the reverse correlation of channel length to Li preference for the two groups implies the existence of optimal dimension around 155–420 nm for the best Li selectivity (Supplementary Fig. 25). Below this dimension, the barrier of sodiation is still low; above this dimension, the barrier of lithiation starts to increase. To validate our prediction, a new platelet particle (Platelet-340 nm) with ~ 340 nm [010] channel length was synthesized (Supplementary Figs. 26-27 and Supplementary Note 2). This particle exhibited the highest Li preference ($0.1 C'\_Li\% = (95.8 \pm 0.3)\%$) with Li to Na selectivity of $2.3 \times 10^4$ and the most significant kinetic barrier difference ($K.B.\_(Li-Na)0.5' = 0.24$ V) was witnessed (Supplementary Fig. 28 and Supplementary Table 10).

Additionally, reversible capacity is also important during lithiation or sodiation (e.g., $Q_{Li}\_1^{st}\_deint$ and $Q_{Na}\_1^{st}\_deint$) for Li extraction performance. Figure 6 shows strong negative trends between the reversible capacity and the [010] channel length or particle size. This suggests that the existence of optimal dimensions of $FePO_4$ particles will also benefit reversibility.

## Discussion

To conclude, a series of particles with varying features were first synthesized and investigated to comprehensively understand the host response upon $Li^+/Na^+$ ion intercalation, aiming to identify the critical features with high Li favorability. For instance, the particles exhibited [010] channel lengths spanning from 20 to 6000 nm, with sizes distributed between $2.5 \times 10^{-4}\ \mu m^3$ and $24\ \mu m^3$.

The diverse electrochemical characteristics observed in these particles, along with the corresponding phase transformation behaviors elucidated through in situ synchrotron XRD, allowed us to categorize the particles into two distinct groups and gather more particle features. Group 1, comprising small particles with [010] lengths below 100 nm, demonstrated structural equilibrium during both lithiation and sodiation transitions. This equilibrium was attributed to fast kinetics and the absence of miscibility and nucleation barriers. In contrast, Group 2, consisting of larger particles with [010] lengths exceeding 500 nm, exhibited a lithiation transition in structural equilibrium but an out-of-equilibrium sodiation transition. This behavior was linked to kinetic and chemo-mechanical barriers hindering sodiation, leading to notable lattice distortions, increased nucleation barrier and coherency strain in the larger particle group.

Consequently, the kinetic and chemo-mechanical overpotential gain of particles in Group 2, results in a higher Li preference during $Li^+$-$Na^+$ co-intercalation. In contrast, the selectivity of small particles was primarily driven by thermodynamic preferences, as their minimal nucleation barrier and faster $Na^+$ diffusion led to a reduction in the kinetic preference for $Li^+$ and considerable non-faradaic ion exchange,

especially at low extraction rates. However, the kinetic lithium preference of small particles can be induced at high currents.

Finally, correlation maps were generated for each group, highlighting the existence of optimal dimensions of $FePO_4$ particles that can be strategically designed to promote both high Li selectivity and reversibility.

## Methods
### Synthesis of FePO_4 particles

A solvothermal synthesis method was used to synthesize all six pristine $LiFePO_4$ particles, each with a slightly different recipe. See Supplementary Note 2 for the detailed synthesis procedure for each particle. After the solvothermal synthesis was completed, all six pristine $LiFePO_4$ particles followed the same washing, carbon coating, and chemical extraction process described in the following to prepare the $FePO_4$ particles for later electrode fabrication.

Specifically, the obtained $LiFePO_4$ precipitates from the solvothermal synthesis were centrifuged three times with deionized water and ethanol, followed by 60 °C drying overnight. To further increase the electronic conductivity of $LiFePO_4$ particles, surface carbon coating is utilized, which has proven to be an effective strategy[7,40,41]. Concretely, the carbon coating procedure involved amalgamating pristine $LiFePO_4$ with sucrose (as the carbon source) in a mass ratio of 5:1 ($LiFePO_4$:sucrose), all while preserving the integrity of the primary particles. The mixture was initially calcinated in an Ar atmosphere at 200 °C for 0.5 h and then heated to 550 °C for 2.5 h. The heating rate is 3 °C $min^{-1}$.

For the chemical extraction of Li from carbon-coated $LiFePO_4$, an oxidizing solution was prepared by dissolving 1.7 g of nitronium tetrafluoroborate ($NO_2BF_4$) in 100 mL of acetonitrile. 1.0 g of carbon-coated $LiFePO_4$ powder was immersed into the solution and stirred for 24 h at room temperature (20 - 25 °C). The powder was then washed several times with acetonitrile and finally dried in a vacuum oven for 12 h at 60 °C. Finally, we will have $FePO_4$ particles ready for use.

### Preparation of electrodes

All $FePO_4$ electrodes were prepared by casting a slurry of $FePO_4$, Super P carbon black (MTI Corporation; Item Number: Lib-SP; average particle size ~ 40 nm; purity ≥ 99.5%), and polyvinylidene fluoride (MTI Corporation; Item Number: Lib-PVDF; purity ≥ 99.5%) with a mass ratio of 80:10:10, in N-methyl-2-pyrrolidone. The low mass loading working electrodes (~2.5 mg/cm²) were prepared by drop casting the slurry on a $0.5 \times 0.5$ cm² geometrical surface of a carbon paper (TGP-H-060, Fuel Cell Earth, 190 μm in thickness, 78% porosity) current collector of $2.5 \times 0.5$ cm². Besides, to increase the hydrophilicity of the carbon paper, the $0.5 \times 0.5$ cm² drop-casting area was cleaned with argon plasma at 100 watts for 1 minute before the drop-casting of the slurry. $FePO_4$ counter electrodes were made with the same slurry depositing on carbon felt (Alfa Aesar) disks (0.9525 cm diameter × 3.18 mm thickness, around 240 g/m² in areal weight). The active material mass loading on the counter electrodes ranged between 60 and 70 mg cm⁻². Platelet-1200 nm $FePO_4$ electrodes were used to prepare $Li_xFePO_4$/$Na_yFePO_4$ counter electrodes. Specifically, the $FePO_4$ electrodes were galvanostatic lithiation/sodiation in 1 M LiCl(aq)/NaCl(aq) at a C/20 (8.5 mA/g) rate until reaching a −0.6 V versus Ag/AgCl voltage cutoff. The larger mass loading of the counter electrode ensures we have enough ion stock in the counter electrode to avoid side reactions from water splitting or pH fluctuations. C/N describes the current to (de)intercalate the electrode in Nh.

### Electrochemical methods

All electrochemical operations were performed on a Bio-Logic VMP3 workstation using a three-neck round-bottomed flask at room temperature (20 - 25 °C). $N_2$ (purity > 99.998%) was continuously bubbled into the solution to avoid side reactions caused by dissolved $O_2$[4,7].

**Evaluation of the aqueous electrochemical energy storage performance.** To verify the quality and measure the accessible capacity of fabricated $FePO_4$ working electrodes, the working electrodes were cycled in either 60 mL 1 M LiCl aqueous solutions (17 mA/g; paired with $Li_xFePO_4$ counter electrodes) or 60 mL 1 M NaCl aqueous solutions (15.4 mA/g; paired with $Na_yFePO_4$ counter electrodes) between −0.6 V and 0.6 V (vs. Ag/AgCl/KCl (4.0 M)) at room temperature (20 - 25 °C) (Supplementary Fig. 9). As shown in Supplementary Fig. 10, we also tested the cycling performance of the $FePO_4$ electrodes in 60 mL 1 M LiCl aqueous solutions at the elevated C rate (0.5 C, equivalent to 85 mA/g; paired with $Li_xFePO_4$ counter electrodes) between −0.6 V and 0.6 V (vs. Ag/AgCl/KCl (4.0 M)) at room temperature (20 - 25 °C).

**Evaluation of Li extraction performance.** The Li extraction performance of the six particles was examined using 1: 1000 Li: Na molar ratio solutions (1 mM LiCl and 1 M NaCl mixed solution). The 1: 1000 Li: Na ratio is selected based on the compositions of brines and geothermal fluids[7,8]. Prior to $Li^+$-$Na^+$ co-intercalation, electrodes are precycled once in 60 mL 1 M LiCl aqueous solutions (17 mA/g; paired with $Li_xFePO_4$ counter electrodes) between −0.6 V and 0.6 V (vs. Ag/AgCl/KCl (4.0 M)) at room temperature (20 - 25 °C) (Supplementary Fig. 9) to measure the accessible capacity. The calculations of applied current and depth of discharge for the $Li^+$-$Na^+$ co-intercalation and later Li recovery were based on the delivered capacity during the initial delithiation rather than the capacity calculated from the mass. We used C′ instead of C to differentiate the C-rates. For instance, 0.1 C′ for the Platelet-20 nm particle will be 15.9 mA/g. The accessible capacities for all particles are summarized in Supplementary Table 5.

During the $Li^+$-$Na^+$ co-intercalation process, all the working electrodes, paired with $Na_yFePO_4$ counter electrodes, would undergo intercalation in either 500 mL (for high mass loading working electrodes) or 200 mL (for low mass loading working electrodes) of synthetic brine solutions (1 mM LiCl and 1 M NaCl mixed solution or 10 mM LiCl and 1 M NaCl mixed solution) until 70% of the accessible capacity using C′/30, 0.1 C, 0.2 C′, 0.5 C′, 1 C′, 2 C′, 4 C′ or 6 C′ current density. It is worth mentioning that, for Platelet-600 nm particles at 2 C′ co-intercalation in 1: 1000 Li: Na, ~57% accessible capacity was used due to the reach of cutoff voltage (Supplementary Fig. 18c). Similarly, in the case of 6 C′ co-intercalation in a 1: 100 Li: Na solution, ~60% accessible capacity was used due to the reach of cutoff voltage (Fig. 5d).

During the recovery process, after finishing the Li extraction in synthetic brine solutions, the electrode was first rinsed in three fresh 60 mL DI water for 30 min with continuous $N_2$ bubbling to remove excess adsorbed cations. The electrode was then de-intercalated in 30 mM $NH_4HCO_3$ solution with a constant current of C′/30 (e.g., 5.3 mA/g for Platelet-20 nm particle), using a graphite rod (Sigma-Aldrich, 99.995%, 10 cm length × 6 mm diameter) as the counter electrode and Ag/AgCl/KCl (4.0 M) as the reference electrode. Before and after the deintercalation process, the solution was collected for ICP-MS for $Li^+$ and $Na^+$ concentration measurement. We measure $Li^+$ and $Na^+$ concentration in the recovery solution and make sure the total ion amount measured matches the electrochemical deintercalation capacity with ~5% error tolerance.

**Evaluation of non-faradaic ion-exchange behavior.** The non-faradaic ion-exchange behavior of the six particles was examined using 1: 1000 Li: Na molar ratio solutions (1 mM LiCl and 1 M NaCl mixed solution). Similarly, electrodes were precycled once in 60 mL 1 M LiCl aqueous solutions. Various Li-ion pre-intercalated particles using 10, 35, or 50% of the accessible capacity were investigated and labeled as DOD_Li0.1′/0.35′/0.5′. The Li-ion pre-intercalated working electrodes were first rinsed in three fresh 60 mL DI water for 30 min with continuous $N_2$ bubbling to remove excess adsorbed Li ions before soaking in 1: 1000 Li: Na molar ratio solutions. During the open circuit voltage (OCV)

monitoring in the Li-Na mixed solution, $Na_yFePO_4$ was paired as the counter electrode, with Ag/AgCl/KCl (4.0 M) as the reference electrode. Right after the soaking, the electrodes were rinsed in three fresh 60 mL DI water for 30 min with continuous $N_2$ bubbling for further use.

## Indicators for Li extraction performance

Two types of indicators are reported here. One is Li/(Li+Na) or Na/(Li+Na), which denotes the molar ratio of $Li^+$/$Na^+$ in the recovery solution. Another indicator is the Li selectivity, which is defined by the following equation:

$$Li_{selectivity} = \frac{([Li]/[Na])_{final}}{([Li]/[Na])_{initial}} \quad (1)$$

where $([Li]/[Na])_{final}$ is the $Li^+$/$Na^+$ molar ratio in the recovery solution, and $([Li]/[Na])_{initial}$ is the $Li^+$/$Na^+$ molar ratio in the synthetic brine solution.

## X-ray diffraction (XRD) characterization

For in-house measurements of synthesized $LiFePO_4$ and $FePO_4$ powder, XRD was carried out on Rigaku MiniFlex 600 diffractometer, using Cu Kα radiation (Kα 1: 1.54059 Å; Kα 2: 1.54441 Å; Kα 12 ratio: 0.4970). The tube voltage and the current used were 40 kV and 15 mA. Diffractograms were repeated three times to increase the S/N ratio with a 0.02° step width and a 10°/min speed. Rietveld refinement was executed on synthesized pristine $LiFePO_4$ and $FePO_4$ particles using GSAS-II software (Supplementary Figs. 7, 8, and Supplementary Tables 2 and 4). For in-house measurements of carbon cloth or carbon paper electrodes, XRD was carried out on Rigaku SmartLab multi-purpose diffractometer, using Cu Kα radiation (Kα 1: 1.54059 Å; Kα 2: 1.54441 Å; Kα 12 ratio: 0.4970). The tube voltage and the current used were 40 kV and 40 mA. Diffractograms were repeated five times to increase the S/N ratio with a 0.02° step width and a 10°/min speed. In situ and ex situ synchrotron XRD measurements were conducted at 13-BM, 15-ID, and 33-BM[42] beamlines of Advanced Photon Source. A specially designed three-electrode cell was used for in situ measurements, allowing aqueous electrolyte solution to flow across the electrode while changing the current and monitoring the phase transformation by synchrotron. During the lithiation of the electrodes, 1 M LiCl aqueous solutions were used as electrolytes, paired with $Li_xFePO_4$ carbon felt counter electrodes and one leakless miniature Ag/AgCl reference electrode (Edaq Inc, ET072-1). During the sodiation of the electrodes, 1 M NaCl aqueous solutions were used as electrolytes, paired with $Na_yFePO_4$ carbon felt counter electrodes and one leakless miniature Ag/AgCl reference electrode (Edaq Inc, ET072-1). During non-faradaic ion exchange of Platelet-20 nm pre-lithiated particles (DOD_Li0.50′), 1 mM LiCl and 1 M NaCl mixed solutions were used as electrolytes, paired with $Na_yFePO_4$ carbon felt counter electrodes and leakless miniature Ag/AgCl reference electrode (Edaq Inc, ET072-1).

## Inductively coupled plasma-mass spectrometry (ICP-MS) characterization

3% $HNO_{3(aq)}$ was used as the diluting matrix for all the Li recovery solutions. Besides, the non-faradaic ion-exchanged particles were first washed with distilled water 3-5 times, then digested with aqua regia solution for three days to ensure complete dissolution. The resulting supernatant was diluted with 3% $HNO_3$ for later ICP-MS measurement. All the measurements used either Thermo iCAP Q ICP-MS or Thermo iCAP RQ ICP-MS.

## Scanning electron microscopy (SEM) characterization

Scanning electron microscopy (SEM, Zeiss Merlin) was performed at the accelerating voltage of 10 kV.

**Scanning transmission electron microscopy (STEM) characterization**

STEM images were acquired using the aberration-corrected JEOL ARM200CF at the University of Illinois at Chicago. A cold field emission source operated at 200 kV was equipped. The high-angle annular dark-field (HAADF) detector angle was 90-270 mrad to give Z contrast images with a less than 0.8 Å spatial resolution. The low-angle annular dark-field (LAADF) detector angle ranged between 40 and 120 mrad.

## Data availability

The data used in this study are available in the main text and the Supplementary Information. All other data are available from the corresponding author upon request. Source data are provided with this paper.

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

## Acknowledgements

This work was supported by the University of Chicago Materials Research Science and Engineering Center, which is funded by the National Science Foundation under award number DMR-2011854. The X-ray diffraction and STEM characterizations received support from Department of Energy Office of Science (DE-SC0023317). This work made use of instruments in the Electron Microscopy Core, Research Resources Center in University of Illinois at Chicago. Portions of this work were performed at GeoSoilEnviroCARS (The University of Chicago, Sector 13), Advanced Photon Source, Argonne National Laboratory. GeoSoilEnviroCARS is supported by the National Science Foundation-Earth Sciences (EAR-1634415). J.E.S and P.J.E. received further support from Department of Energy-Geosciences (DE-SC0019108). This research used resources of the Advanced Photon Source, a U.S. Department of Energy Office of Science User Facility operated for the DOE Office of Science by Argonne National Laboratory under Contract No. DE-AC02-06CH11357. NSF's ChemMatCARS, Sector 15 at the Advanced Photon Source, Argonne National Laboratory is supported by the Divisions of Chemistry and Materials Research, National Science Foundation, under grant number NSF/CHE-1834750.

## Author contributions

G.Y. and C.L. conceived the idea and designed the experiment. G.Y. performed all the experimental work and analyzed the experimental data. J.W. carried out and interpreted the DFT calculations. E.A. and S.C. assisted with materials synthesis. P.J.E, J.E.S, M.K.B, E.K, and H.Z. assisted with synchrotron XRD measurements. Y.H. assisted with electron diffraction collecting. S.Z. assisted with in situ synchrotron XRD experiments. R.W. assisted with SEM imaging. W.C. supervised the DFT calculations. C.L. supervised the work. G.Y. and C.L. wrote the manuscript, and all authors revised the manuscript.

## Competing interests

The authors declare no competing interests.
