## [Peer Review File · Nature Communications]

Identifying critical features of iron phosphate particle for lithium preferenceREVIEWER COMMENTS

Reviewer #1 (Remarks to the Author):

The manuscript submitted by Professor Chong Liu et al., "Identifying critical features of iron phosphate particle for lithium preference," describes the effects of particle morphologies on the electrochemical performance and selectivity of the material for its application as a selective cathode for lithium recovery/production. The manuscript analyzes the differences in the intercalations of sodium and lithium in the olivine/heterosite structure from the thermodynamic and kinetic perspectives. Minor modifications/clarifications are suggested before its publication since the topic is highly relevant, and the study was performed in great detail.

Page 8 – lines 177-180.

"As shown in Figures 3^a... one group displayed minimal and small voltage hysteresis during (de)lithiation and (de)sodiation..."

There is a higher polarization in sodium cells than in lithium cells. The sodium intercalation is polarized down to 0.0 V, but it is 0.2 V for lithium.

In the same figure, 3b, It is surprising that the 1200 nm sample breaks the tendency. The reduction potential is clearly higher than for 600 nm sample. Any explanation for this behavior?

On the other hand, it is challenging to follow the graphs with so many similar colors; maybe introducing a marker for each sample could help.

Page 8 – lines 191 – 194

"It is worth highlighting that the Platelet-600 nm particle in Group 2 has the largest difference in cycling features between (de)lithiation and (de)sodiation. Its size is relatively small to release strain penalty and facilitate fast Li⁺ (de)intercalation kinetics but is large enough to induce kinetic and chemo-mechanical barriers during Na⁺ (de)intercalation."

What is the meaning of the largest differences? The whole group 2 has substantial differences between sodium and lithium intercalation. Can the authors clarify these sentences, please?

Figure 3. The color associated with the samples' intercalation in lithium cannot be differentiated.

The discussion about the XRD in-situ evolution upon lithium/sodium intercalations is very complete; however, similar studies have been previously published, yet there is no reference in this section probing that the results obtained are similar to those already done.

The author indicates that the in-situ study of group 2 of particles was done at a lower scan rate ($\approx C/100$) because of the worse rate capability; however, the cycling behavior shown in Figure 3b at C/10 is not so bad. The change in the C-rate is enormous and could affect the performance of the material.

Page 13 - Line 300

Why did the authors select 70% accessible capacity for the Li extraction experiments? Did the polarization start at higher DOD?

Page 13 – Lines 303 – 316

The study of higher mass loading seems confusing and does not contribute much to the goals of the manuscript. It could be removed. In any case, the profile's potential vs. capacities should be shown if the discussion is maintained; they are needed to analyze the conclusions obtained by the authors.

The abbreviation used in Figure 6 for the correlation coefficient maps should be included to help read

the results obtained.

Reviewer #2 (Remarks to the Author):

Yan et al presented a comprehensive study of phase separation and solid solution upon electrochemical lithium and sodium intercalation into the AFePO₄ olivine structure (A = Li, Na), and aimed to correlate such results with electrochemical lithium extraction. The authors did an excellent job studying six different particle sizes and morphologies, controlled using solvothermal synthesis conditions. It is my opinion that either half of this paper (phase evolution of olivine or selective lithium extraction) is a significant accomplishment, and the combination of both in one paper makes it very novel and impactful.

However, it appears that some inconsistencies between the two halves of the paper should be clarified. In particular, the authors claim that the higher over potential of Na in larger (Group 2) particles is responsible for the greater degree of lithium selectivity. However, it also seems like the overpotential in smaller particles (Group 1) is essentially negligible. While the selectivity for Li is only about 35-60% (Fig. 5a), that occurs in a solution with >1000:1 Na:Li ratio. Can the authors clarify why even the small particles, where the over potentials in Fig. 3 are similar, result in an extremely high lithium selectivity when accounting for the ions in solution?

In addition, I think this work, due to its length and comprehensiveness, would benefit from short "sectional summaries", as well as better transitions between the different section. The discussion section could also become longer to tie together the various parts of this work. This will allow the reader to better follow the work.

Finally, it was not clear if Figure 1 was based on previous research, or if it represents the results in this paper. The references were partly discussed in the introduction paragraph. If these were based on previous work, it could help if the reference numbers could be embedded into the figure next to each operational regime (e. g. Group 1, SS Transition, Li; Group 2, Phase Separation, Li; etc). The phase evolution not observed could then be highlighted without a reference.

Response to reviewers' comments:

We would like to thank the reviewers for their efforts in reviewing our manuscript. We have revised our manuscript accordingly. Below please find our point-by-point response to the comments received. The major changes in our revised manuscript have been marked in Red.

Reviewer #1:

The manuscript submitted by Professor Chong Liu et al., "Identifying critical features of iron phosphate particle for lithium preference," describes the effects of particle morphologies on the electrochemical performance and selectivity of the material for its application as a selective cathode for lithium recovery/production. The manuscript analyzes the differences in the intercalations of sodium and lithium in the olivine/heterosite structure from the thermodynamic and kinetic perspectives. Minor modifications/clarifications are suggested before its publication since the topic is highly relevant, and the study was performed in great detail.

We appreciate the positive comments and valuable suggestions from the reviewer. Below we provided point-to-point replies to the reviewer's questions.

Page 8 – lines 177-180.

"As shown in Figures 3^a... one group displayed minimal and small voltage hysteresis during (de)lithiation and (de)sodiation...".

There is a higher polarization in sodium cells than in lithium cells. The sodium intercalation is polarized down to 0.0 V, but it is 0.2 V for lithium.

In the same figure, 3b, It is surprising that the 1200 nm sample breaks the tendency. The reduction potential is clearly higher than for 600 nm sample. Any explanation for this behavior?

We thank the reviewer for the great question. We observed that during sodiation, as shown in Figs. 4g-j, especially Supplementary Figs. 16c-d and Supplementary Fig. 17 in the revised manuscript, although both 600 nm and 1200 nm particles experienced strong lattice distortion during Na⁺ intercalation without distinct phase transformations, we can still see the emerging of Na_{2/3}FePO₄ intermediate phase in the 1200 nm particles during operando sodiation to help release the chemo-mechanic barrier and release the volumetric strain. We think the higher (020) facet exposure ratio and more accessible storage sites (the (010) facet area to [010] channel length ratio) facilitate the phase behavior of 1200 nm particles compared with 600 nm particles.

We highlighted the related contents in the paper as:

“More specifically, compared to Platelet-600 nm particles, Platelet-1200 nm particles already started to transit to Na_{2/3}FePO₄ phase at the late stage of in situ sodiation before relaxation (Supplementary Fig. 16d),

which demonstrates better capability to release the chemo-mechanical strain and also consistent with the observed higher sodiation voltage (Fig. 3d).”

On the other hand, it is challenging to follow the graphs with so many similar colors; maybe introducing a marker for each sample could help.

We thank the reviewer for the comment. We modified the Figure 3 in the paper as:

Figure 3 Electrochemical response during lithiation or sodiation. a-b, 1st electrochemical cycle under

17 mA/g (equivalent to 0.1C based on theoretical capacity of LiFePO_4) in 60 ml 1M LiCl aqueous solution (paired with Ag/AgCl/KCl (4.0 M) reference and Li_xFePO_4 counter electrodes). **c-d**, 1st electrochemical cycle under 15.4 mA/g (equivalent to 0.1C based on theoretical capacity of NaFePO_4) in 60 ml 1M NaCl aqueous solution (paired with Ag/AgCl/KCl (4.0 M) reference and Na_yFePO_4 counter electrodes). **e**, Bar chart comparisons of end potential collected right after different depth-of-discharge (DOD) in 60 ml 1M LiCl (left panel) or NaCl (right panel) aqueous solution, which corresponds to the value at the bottom of the bar, and open-circuit potential after 20 hours of relaxation without currents, corresponding to the value at the top of the bar. See Methods for electrode preparation and DOD calculation. The dashed lines denotes the calculated thermodynamic voltage for specific reactions (See Supplementary Note 1 for computation details).

Page 8 – lines 191 – 194

"It is worth highlighting that the Platelet-600 nm particle in Group 2 has the largest difference in cycling features between (de)lithiation and (de)sodiation. Its size is relatively small to release strain penalty and facilitate fast Li^+ (de)intercalation kinetics but is large enough to induce kinetic and chemo-mechanical barriers during Na^+ (de)intercalation."

What is the meaning of the largest differences? The whole group 2 has substantial differences between sodium and lithium intercalation. Can the authors clarify these sentences, please?

We thank the reviewer for the question. We agreed with the reviewer that the whole Group 2 has substantial differences compared with particles in Group 1. What we want to further emphasize here is the better rate capability during (de)lithiation (Supplementary Figure 10) but still low sodiation voltage (high sodiation overpotential) of Platelet-600 nm particles (Figure 3d). We have modified the context to better clarify our claims:

"It is worth highlighting that the Platelet-600 nm particle in Group 2 has the largest difference in cycling features between (de)lithiation and (de)sodiation. Platelet-600 nm demonstrate better (de)lithiation rate capability; however, both Platelet-1200 nm and Cuboid-6000 nm particles displayed significant capacity decay during (de)lithiation at 0.5C (Supplementary Figure 10). In summary, the size of the Platelet-600 nm particle is relatively small to release strain penalty and facilitate fast Li^+ (de)intercalation kinetics but is large enough to induce kinetic and chemo-mechanical barriers during Na^+ (de)intercalation."

Figure 3. The color associated with the samples' intercalation in lithium cannot be differentiated.

We thank the reviewer for the comment. We have modified the Figure 3 in the manuscript as shown above.

The discussion about the XRD in-situ evolution upon lithium/sodium intercalations is very complete; however, similar studies have been previously published, yet there is no reference in this section probing that the results obtained are similar to those already done.

We thank the reviewer for the great comment. Indeed, many great works have studied phase evolutions of (de)lithiation of Li_xFePO_4 with in situ XRD and attention was paid to in situ (de)sodiation. Previously, we

mainly cited related works when discussing Figure 1. It is a great suggestion to add previous work during the discussion of our *in situ* synchrotron XRD data. We have modified the “*Particle morphology dependent phase evolutions during lithiation or sodiation*” section and cited the following related works:

References:

1. Wagemaker, M. et al. Dynamic Solubility Limits in Nanosized Olivine LiFePO₄. Journal of the American Chemical Society 133, 10222-10228, doi:10.1021/ja2026213 (2011).
2. Meethong, N., Huang, H.-Y. S., Carter, W. C. & Chiang, Y.-M. Size-Dependent Lithium Miscibility Gap in Nanoscale Li_{1-x}FePO₄. Electrochemical and Solid-State Letters 10, A134, doi:10.1149/1.2710960 (2007).
3. Lu, J. C., Chung, S. C., Nishimura, S. & Yamada, A. Phase Diagram of Olivine Na_xFePO₄ (0 < x < 1). Chemistry of Materials 25, 4557-4565, doi:10.1021/cm402617b (2013).
4. Delacourt, C., Poizot, P., Tarascon, J.-M. & Masquelier, C. The existence of a temperature-driven solid solution in Li_xFePO₄ for 0 ≤ x ≤ 1. Nat. Mater. 4, 254-260, doi:10.1038/nmat1335 (2005).
5. Delmas, C., Maccario, M., Croguennec, L., Le Cras, F. & Weill, F. Lithium deintercalation in LiFePO₄ nanoparticles via a domino-cascade model. Nat. Mater. 7, 665-671, doi:10.1038/nmat2230 (2008).
6. Hess, M., Sasaki, T., Villevieille, C. & Novák, P. Combined operando X-ray diffraction–electrochemical impedance spectroscopy detecting solid solution reactions of LiFePO₄ in batteries. Nature Communications 6, 8169, doi:10.1038/ncomms9169 (2015).
7. Yamada, A. et al. Room-temperature miscibility gap in Li_xFePO₄. Nat. Mater. 5, 357-360, doi:10.1038/nmat1634 (2006).
8. Liu, H. et al. Capturing metastable structures during high-rate cycling of LiFePO₄ nanoparticle electrodes. Science 344, 7, doi:10.1126/science.1252817 (2014).

The author indicates that the *in-situ* study of group 2 of particles was done at a lower scan rate ($\approx C/100$) because of the worse rate capability; however, the cycling behavior shown in Figure 3b at C/10 is not so bad. The change in the C-rate is enormous and could affect the performance of the material.

We thank the reviewer for the comment. We need to clarify that during the *in situ* investigation of phase evolutions of particles in Group 2 (Figures 4e-h in the manuscript), we were using $\sim 0.1C$ lithiation/sodiation rate for both Platelet-600 nm (0.092C lithiation and 0.090C sodiation) and Platelet 1200 nm particles (0.077C lithiation and 0.086C sodiation).

Page 13 - Line 300

Why did the authors select 70% accessible capacity for the Li extraction experiments? Did the polarization start at higher DOD?

We thank the reviewer for the great question. Our accessible capacity is determined by the cycling capacity in LiCl (aq). In a mixed LiCl: NaCl solution with a co-intercalation process, the total capacity could decrease due to Na⁺ co-intercalation. Also, the higher polarization at the end of intercalation will increase the chance of Na⁺ intercalation. Therefore, we used 70% accessible capacity during the (de)lithiation

process for the later Li extraction experiments and did not push to use the full capacity based on Li⁺ (de)intercalation.

Page 13 – Lines 303 – 316

The study of higher mass loading seems confusing and does not contribute much to the goals of the manuscript. It could be removed. In any case, the profile's potential vs. capacities should be shown if the discussion is maintained; they are needed to analyze the conclusions obtained by the authors.

We agree with the reviewer that the discussion of mass loading is probably redundant. We were trying to emphasize the importance of electrode engineering for future practical applications, which is indeed not the primary goal of this project. We have removed the discussion of this part.

The abbreviation used in Figure 6 for the correlation coefficient maps should be included to help read the results obtained.

We thank the reviewer for the comment. We added the descriptions into the figure captions to help the reading of the figure. We also summarized the definition of each variable and values for each particle in Supplementary Tables 8, 9, and Supplementary Note 3. We have modified the caption of Figure 6 in the manuscript:

“...Specifically, " $Q_{Li/Na}$ " denotes the delivered capacity in the first de-lithiation/sodiation process; " $K.B.$ " denotes the measured kinetic barrier/potential change during relaxation; " $0.1C'_E$ " denotes the end potential after 0.1C' constant current lithiation/sodiation; " $R.E.$ " denotes the end potential after relaxation; See Supplementary Tables 8, 9, and Supplementary Note 3 for the complete definition of each variable or summary of the values for each particle.”

Reviewer #2:

Yan et al presented a comprehensive study of phase separation and solid solution upon electrochemical lithium and sodium intercalation into the AFePO₄ olivine structure (A = Li, Na), and aimed to correlate such results with electrochemical lithium extraction. The authors did an excellent job studying six different particle sizes and morphologies, controlled using solvothermal synthesis conditions. It is my opinion that either half of this paper (phase evolution of olivine or selective lithium extraction) is a significant accomplishment, and the combination of both in one paper makes it very novel and impactful.

We appreciate the highly positive comments from the reviewer. Below we provided point-to-point replies to the reviewer's questions.

However, it appears that some inconsistencies between the two halves of the paper should be clarified. In particular, the authors claim that the higher over potential of Na in larger (Group 2) particles is responsible for the greater degree of lithium selectivity. However, it also seems like the overpotential in smaller particles (Group 1) is essentially negligible. While the selectivity for Li is only about 35-60% (Fig. 5a), that occurs in a solution with >1000:1 Na:Li ratio. Can the authors clarify why even the small particles, where the over potentials in Fig. 3 are similar, result in an extremely high lithium selectivity when accounting for the ions in solution?

We thank the reviewer for the great question. This is due to the intrinsic Li preference of the FePO₄ hosts. As we can see from Figures 3a, 3c, and 3e in the revised manuscript, the sodiation voltage (~ 0 V vs. Ag/AgCl) and lithiation (~ 0.2 V vs. Ag/AgCl) voltage difference is ~ 0.2V for particles in Group 1. In a 1mM: 1M Li: Na mixed solution, considering the potential shift from concentration correction, the lithiation potential will shift to ~ 0.02 V vs. Ag/AgCl ($0.2 + 0.0592 \cdot \log[1 \text{ mM}]/[1 \text{ M}] = 0.02 \text{ vs. Ag/AgCl}$), which is close to the sodiation voltage, enabling a comparable intercalation preference between Li and Na.

Therefore, the thermodynamic preference of FePO₄ provides a baseline of Li preference while the particle features are critical to induce different kinetic pathways and barrier energies, which can be harnessed to enhance the Li to Na selectivity.

In addition, I think this work, due to its length and comprehensiveness, would benefit from short "sectional summaries", as well as better transitions between the different section. The discussion section could also become longer to tie together the various parts of this work. This will allow the reader to better follow the work.

We appreciate the reviewer's comment and agree that our work could benefit from brief summaries, better transitions, and more detailed discussions.

At the end of "*Quantification of particle morphology features and electrochemical response during lithiation or sodiation*" section we added:

“This rich collection of particle morphology characteristics and electrochemical responses will be used to identify critical features associated with high Li preference.”

At the beginning of “*Particle morphology dependent phase evolutions during lithiation or sodiation*” section we highlighted:

“The intercalation pathways and the associated phase evolutions or ion storage mechanisms are also critical in determining the energy barriers for both Li⁺ and Na⁺ intercalation, consequently influencing Li preference⁷.”

At the end of “*Particle morphology dependent phase evolutions during lithiation or sodiation*” section we highlighted:

“The observed phase evolutions confirm the rationale behind the grouping of particles based on their morphological form factors that particles in Group 1 have SS lithiation evolution pathway (in equilibrium) paired with two-stage phase evolution pathway during sodiation (in equilibrium), while particles in Group 2 have phase-separation-dominated lithiation evolutions (in equilibrium) paired with out-of-equilibrium sodiation transition.”

We have also rewritten the discussion section as follows:

“To conclude, a series of particles with varying features were synthesized and investigated to comprehensively understand the host response upon Li⁺/Na⁺ ion intercalation, aiming to identify the critical features with high Li favorability. For instance, the particles exhibited [010] channel lengths spanning from 20 nm to 6000 nm, with sizes distributed between $2.5 \times 10^{-4} \mu\text{m}^3$ and $24 \mu\text{m}^3$.

The diverse electrochemical characteristics observed in these particles, along with the corresponding phase transformation behaviors elucidated through *in situ* synchrotron XRD, allowed us to categorize the particles into two distinct groups and gather more particle features. Group 1, comprising small particles with [010] lengths below 100 nm, demonstrated structural equilibrium during both lithiation and sodiation transitions. This equilibrium was attributed to fast kinetics and the absence of miscibility and nucleation barriers. In contrast, Group 2, consisting of larger particles with [010] lengths exceeding 500 nm, exhibited a lithiation transition in structural equilibrium but an out-of-equilibrium sodiation transition. This behavior was linked to kinetic and chemo-mechanical barriers hindering sodiation, leading to notable lattice distortions, increased nucleation barrier and coherency strain in the larger particle group.

Consequently, the kinetic and chemo-mechanical overpotential gain of particles in Group 2, results in a higher Li preference during Li⁺-Na⁺ co-intercalation. In contrast, the selectivity of small particles was primarily driven by thermodynamic preferences, as their minimal nucleation barrier and faster Na⁺ diffusion led to a reduction in the kinetic preference for Li⁺ and considerable non-faradaic ion exchange, especially at low extraction rates. However, the kinetic lithium preference of small particles can be induced at high currents.

Finally, correlation maps were generated for each group, highlighting the existence of optimal dimensions

of FePO₄ particles that can be strategically designed to promote both high Li selectivity and reversibility.”

Finally, it was not clear if Figure 1 was based on previous research, or if it represents the results in this paper. The references were partly discussed in the introduction paragraph. If these were based on previous work, it could help if the reference numbers could be embedded into the figure next to each operational regime (e. g. Group 1, SS Transition, Li; Group 2, Phase Separation, Li; etc). The phase evolution not observed could then be highlighted without a reference.

We appreciate the reviewer for the comment. Figure 1 is introduced and discussed in the introduction paragraph. The concept of grouping based on particle features and kinetic pathways is generated by our group. But indeed, some of these phase evolution pathways have been reported by other work as referenced. We added the previous work reporting different kinetic pathways in the figure caption.

We have modified the caption of Figure 1 in the manuscript as below. It is also worth mentioning that, to our knowledge, the *SS (out of structural equilibrium) transition* during sodiation is first observed in this work.

Figure 1 Schematic illustrations depicting the particle size dependent phase evolutions of olivine FePO_4 particles during lithiation or sodiation. Different color codes denote different phases during lithiation or sodiation. Here, we grouped the particles based on their different phase evolution pathways upon lithiation and sodiation. Some previous works also witnessed some phase transformations, including solid solution (SS) transition during lithiation^{20-22,28}, phase separation transition during lithiation^{25,26}, SS transition out of structural equilibrium during lithiation²⁷⁻³¹, and two-stage sodiation transition (phase separation + SS transition)³². In this work, we observed SS (out of structural equilibrium) transition upon sodiation. The dashed box in the diagram indicates the equilibrium SS transition throughout the range upon sodiation has not been observed experimentally.

REVIEWERS' COMMENTS

Reviewer #1 (Remarks to the Author):

The authors have answered correctly to all my comments and suggestions. In my opinion, the manuscript is ready for its publication. I want to thank the authors for their clarifications and congratulate them for their excellent work.

Reviewer #2 (Remarks to the Author):

All comments have been adequately addressed.

Response to reviewers' comments:

We would like to thank the reviewers again for their efforts in reviewing our manuscript. The changes required by the editorial requests have been marked in Red in our revised manuscript.

Reviewer #1:

The authors have answered correctly to all my comments and suggestions. In my opinion, the manuscript is ready for its publication. I want to thank the authors for their clarifications and congratulate them for their excellent work.

We thank the reviewer for the valuable suggestions and highly positive feedback on our work.

Reviewer #2:

All comments have been adequately addressed.

We appreciate the reviewer's positive comments and valuable suggestions for improving the work.